# Paste Backfill Corrosion Mechanisms in Chloride and Sulfate Environments

**Guangzheng Xu, Kegong Fan, Kun Wang \***  **and Jianguo Ning \***

College of Energy and Mining Engineering, Shandong University of Science and Technology,
Qingdao 266590, China; xuguangzheng98@126.com (G.X.); fankeg@163.com (K.F.)
\* Correspondence: wkun@sdust.edu.cn (K.W.); njglxh@126.com (J.N.)

**Abstract:** To study paste backfill corrosion mechanisms in chloride and sulfate environments, we studied the effect of chloride and sulfate on the strength of paste backfill after 7, 14, 28, and 40 days. The chloride solutions and sulfate solutions in concentrations are 0 g/L, 0.5 g/L, 1.5 g/L, 4.5 g/L, or 15 g/L. The obtained specimens were analyzed by performing uniaxial compressive strength tests, X-ray diffraction (XRD), and scanning electron microscopy (SEM). The results show that chloride and sulfate significantly increased the uniaxial compressive strength of the specimen at a very fast speed in the early stage of the test, and the original structure of the specimen was destroyed and its uniaxial compressive strength decreased with the gradual corrosion. The reason for this characteristic is because the chloride reacts with the paste backfill to form calcium chloroamine hydrate ($Ca_4Al_2O_6Cl_2 \cdot 10H_2O$), and the sulfate reacts with the paste backfill to form dihydrate gypsum ($CaSO_4 \cdot 2H_2O$), mirabilite, and ettringite. In the early stage, these substances can fill the pores to improve the compressive strength, and then expand to damage the structure of the backfill and reduce its compressive strength. In addition, sulfate can enhance the decomposition of C-S-H, which results in a faster destruction of specimens than in chloride environments.

**Keywords:** mine backfill; chloride solution; sulfate solution; corrosion mechanism

## 1. Introduction

Paste backfill is a type of cementitious mine filling material mainly composed of cement, fly ash, and coal gangue that has been widely used worldwide [1–4]. The coal gangue is a kind of solid waste from coal mining and washing, accounting for approximately 10~15% of the total coal production [5]. The continuous accumulation of coal gangue without adequate storage and disposal facilities could lead to human health and environmental disasters in the long run [6–9]. Backfilling with coal gangue as material can effectively reduce the discharge of solid waste and contribute to the sustainable development of the mine [10–13]. The stability of the backfill is an extremely important factor in mine backfilling [14–18], and researchers have focused on the factors affecting backfill strength [19–21].

In recent years, extensive research on the effect of $Cl^-$ and $SO_4^{2-}$ on the strength of mine backfill have been conducted. Wu et al. studied the seepage characteristics of colloidal sand backfill and concluded that the cement hydration reaction inside the backfill structure caused a considerable quantity of hydration products to fill the internal pore structure [22]. Kitazume et al. conducted laboratory tests to determine how the long-term strength of cement-reinforced soil changes with age, concluding that cement-reinforced soil submerged in fresh water and seawater deteriorates sequentially from the outside to the inside with increasing submersion time [23]. Bai et al. studied the influence of acid- and alkali-containing groundwater environments on the strength of cementitious soil, concluding that the acidic environment had a strong corrosive effect, reducing the compressive strength of the soil, whereas the alkaline environment promoted the hydration of the cement in the soil,

thereby improving its compressive strength [24]. Chew et al. found needle-like crystals, namely ettringite, in concrete corroded by sulfate [25]. Chen et al. studied the effects of chloride as an early strength agent on the mechanical properties and microstructure of gangue-cemented paste backfill (GCPB) and found that chloride significantly affects the early-age strength of GCPB [26]. These works are of great significance to understanding the influence of chloride and sulfate on the stability of backfill. In actual implementation, the stability of the backfill will face the threat of mine water after it is injected into the goaf [27,28]. The strength of the backfill is inevitably affected by $Cl^-$ and $SO_4^{2-}$ in the mine water [29–32]. However, the corrosion mechanism of backfill by $Cl^-$ and $SO_4^{2-}$ in mine water environment needs to be further studied. It is important to study the corrosion mechanism of backfill by $Cl^-$ and $SO_4^{2-}$ on the stability of the backfill. Therefore, in this study, the uniaxial compressive strengths and external surfaces of paste backfill specimens exposed to chloride or sulfate solutions with approximate mine water concentrations were accordingly analyzed and tested. The microscopic phase changes and structural changes of the specimens were then investigated to explore the corrosion mechanism of mine water on cementitious paste backfill.

## 2. Experimental Materials and Methods

The paste backfill used in this study was mainly prepared by mixing cement, fly ash and coal gangue. The cement was 42.5# Ordinary Portland Cement (OPC); the first-grade fly ash was produced by Yuanheng Water Purification Material Factory in Gongyi City. Fly ash is mainly composed of a small amount of unburned carbon particles, fine crystals and glass. The addition of fly ash can significantly improve the pore structure of cement paste. The coal gangue was from Bayangol in Inner Mongolia. Its main components are $Al_2O_3$ and $SiO_2$. It also contains a small amount of $Fe_2O_3$, CaO, MgO, $Na_2O$, and $K_2O$. The uniaxial compressive strength of coal gangue used in this study is 21 MPa. The mass ratio of coal gangue cementing filling material is water:fly ash:coal gangue:OPC = 0.5:0.4:9:1. The diameter of crushed coal gangue particles is less than 25 mm and gangue gradation (content of coal gangue with particle diameter < 5 mm) = 0.4. This recipe has been successfully applied to some mines by us.

We selected 100 mm* 100 mm* 100 mm cube mold to make the test piece. To mimic what occurs in real mines, the paste backfill specimens were submerged without special curing in distilled water or a chloride (NaCl) or sulfate ($Na_2SO_4$) solution after demolding. Affected by geological conditions and other factors in China, the dispersion of chloride and sulfate concentrations in various regions is very large [33–35]. Table 1 shows the chloride and sulfate concentrations in some coal mines we collected. Based on the existing work experience [26,36], the final concentration of each are 0.5 g/L, 1.5 g/L, 4.5 g/L, and 15 g/L (NaCl: 0.0086 mol/L, 0.0256 mol/L, 0.0769 mol/L,0.2564 mol/L. $Na_2SO_4$: 0.0035 mol/L, 0.0106 mol/L, 0.0317 mol/L, 0.1056 mol/L).

**Table 1.** Part of mine salt ion content data.

| Name of Coal Mine | Chloride (mg/L) | Sulfate (mg/L) |
|---|---|---|
| Xinzhuangzi | 39.99 | 327.93 |
| Kongji | 115.16 | 43.16 |
| Panyi | 889.72 | 198.03 |
| Wusu | 412.7 | 1026.3 |
| Zhangji | 259.4 | 472 |
| Shuanggou | 182 | 882.01 |
| Linhuan | 213 | 1580.6 |
| Quangou | 107.7 | 1094.5 |
| Lingxin | 1193.6 | 884.3 |
| Haizi | 146.0 | 1028.0 |
| Bulianta | 235.19 | 227.32 |

After 7 d, 14 d, 28 d, and 40 d submersion time, the uniaxial compressive strength of each backfill test block was determined using an RLJW-2000 rock servo testing machine at a loading rate of 0.15 mm/s without lateral restraint. In addition, the changes in the appearance of the test specimens were observed for each submersion time [37]. The D/Max2500PC X-ray diffractometer produced by Rigaku Co., Ltd. was used for microstructural analysis. The diffraction angle range used was 10–90°, the data point interval was 0.02°, and the scanning step length was 4°/min [38]. The Nova Nano SEM450 high-resolution scanning electron microscope (SEM) produced by FEI was used to observe the microstructure of the specimens at a resolution of 1 nm (at 15 kV) to 1.8 nm (at 3 kV, Helix detector). Each sample was sprayed with gold prior to SEM observation [39–41].

## 3. Results and Analysis

### 3.1. Effect of Chloride and Sulfate Solutions on the External Surface of the Specimens

Understanding the outer surface changes of the backfill paste specimens when subjected to different corrosive environments forms the basis for studying the corrosion mechanisms of $Cl^-$ and $SO_4^{2-}$. The outer surfaces of the backfill paste specimens submerged for 28 days in different concentrations of chloride and sulfate solutions are accordingly shown in Figure 1.

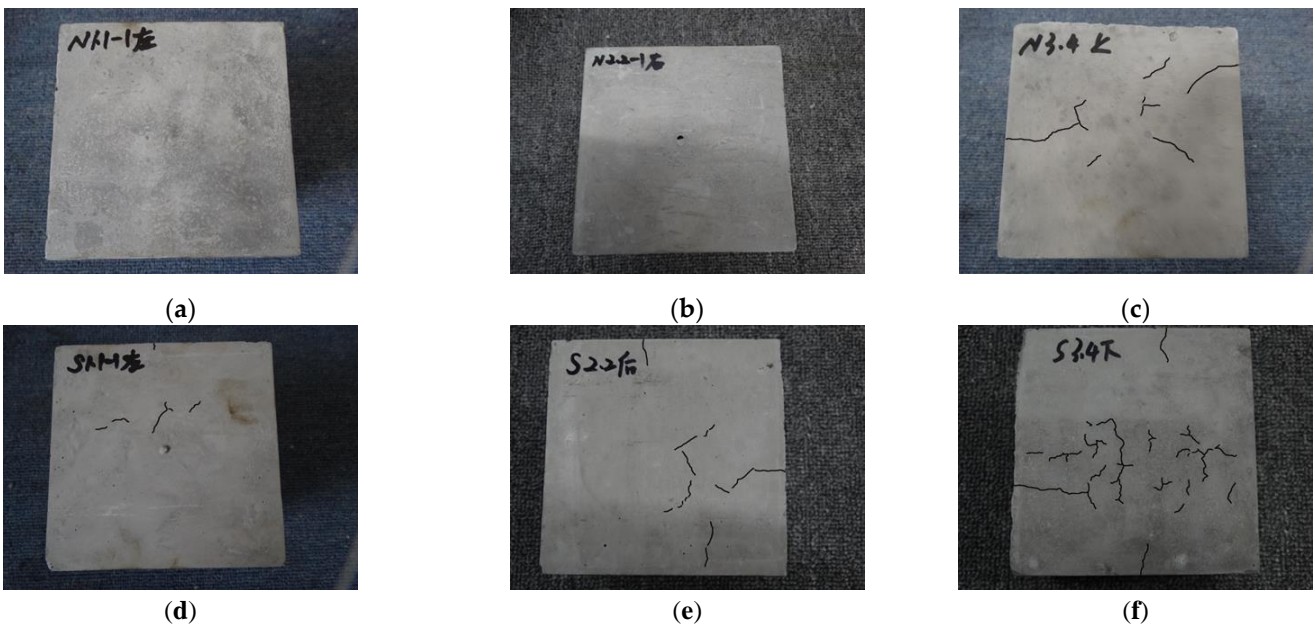

|  |  |  |
|:--:|:--:|:--:|
| (**a**) | (**b**) | (**c**) |
| (**d**) | (**e**) | (**f**) |

**Figure 1.** Appearances of specimens submerged in (**a**) 0.5 g/L (**b**) 4.5 g/L, (**c**) 15 g/L chloride solution, (**d**) 0.5 g/L, (**e**) 4.5 g/L, and (**f**) 15 g/L sulfate solution for 28 days (the size of the test piece face facing the photo is 100 mm*100 mm).

It can be seen from Figure 1 that the outer surface of the specimens submerged in different concentrations of chloride solution exhibited basically no changes, with only a few cracks appearing in the specimen submerged in the highest-concentration solution. However, as the concentration of the sulfate solution increased, an increasing quantity of microcracks appeared on the outer surface of the specimen (as shown by the black lines in the figure). Thus, the macroscopic observation of the outer surface of the test specimens indicated that the corrosion of $Cl^-$ on the paste backfill specimen was much weaker than that of $SO_4^{2-}$ in 28 days, and that the higher the concentration of $SO_4^{2-}$, the greater the degree of influence on the specimen.

*3.2. Effect of Chloride and Sulfate Solutions on the Compressive Strength of the Specimens*

3.2.1. Effect of $Cl^-$ Corrosion

Figure 2 shows the uniaxial compressive strengths of the backfill paste specimens after 7 d, 14 d, 28 d, and 40 d soaking in distilled water or 0.5 g/L, 1.5 g/L, 4.5 g/L, or 15 g/L chloride solutions.

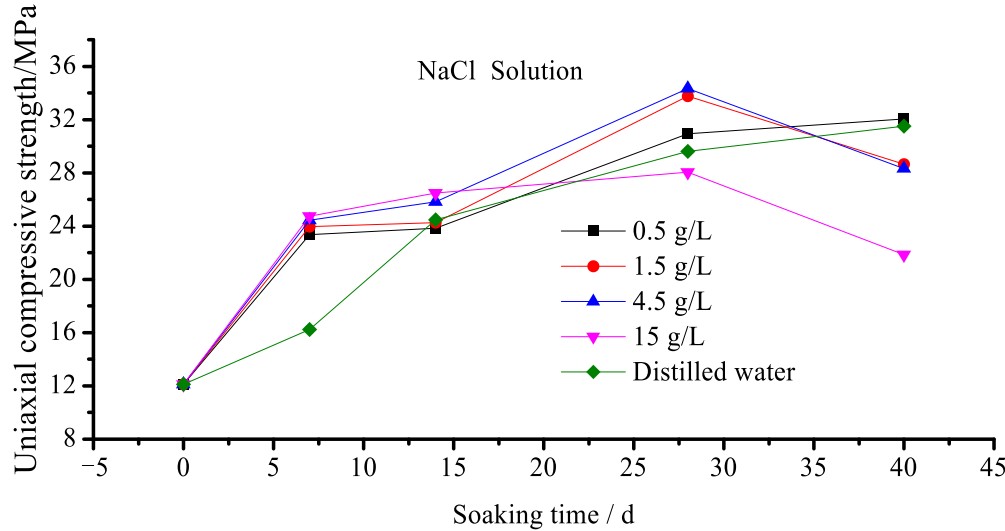

**Figure 2.** Uniaxial compressive strength of paste backfill specimens when submerged in different concentrations of chloride solution according to soaking time.

It can be seen from Figure 2 that the uniaxial compressive strengths of the specimens were considerably enhanced in the early stage of the experiment. Furthermore, the uniaxial compressive strengths of the specimens submerged in a chloride solution increased much faster than that of the specimen submerged in distilled water, and the rate of increase in compressive strength was greater for the specimens in higher-concentration chloride solutions than for those in lower-concentration chloride solutions. In the later stage of the experiment, the uniaxial compressive strengths of the specimens in higher-concentration chloride solution began to gradually decrease, and the higher the solution concentration, the faster the decrease. As the paste backfill had a loose and porous structure, the observed increase in uniaxial compressive strength can be primarily attributed to the generation of hydration products that fill the internal voids in the specimen during the ongoing hydration reaction, making the specimen more compact. Thus, based on comparison with the specimens cured in distilled water and lower-concentration chloride solutions, the reason for the rapid increase in uniaxial compressive strength in the early stage of the experiment may be that the $Cl^-$ participated in curing the paste. Aiding the hydration process increased the rate of hydration and generated substances that filled the pore structure of the test specimen, increasing its uniaxial compressive strength faster in the high-concentration chloride solution than in the low-concentration chloride solution or distilled water. The uniaxial compressive strengths of the specimens submerged in the chloride solution decreased over time because as their ages increased, more and more hydration products were formed, continually filling the pore structure; however, such products often swell, expanding the pores and applying pressure on the pore walls, resulting in the generation of tensile stress. When this tensile stress exceeded the material limit, fractures were induced causing microcracks to appear, reducing the strength of the paste backfill.

3.2.2. Effect of $SO_4^{2-}$ Corrosion

The uniaxial compressive strength of the paste backfill specimen after submersion in distilled water or sulfate solutions with concentrations of 0.5 g/L, 1.5 g/L, 4.5 g/L, or 15 g/L for 7 d, 14 d, 28 d, and 40 d is shown in Figure 3.

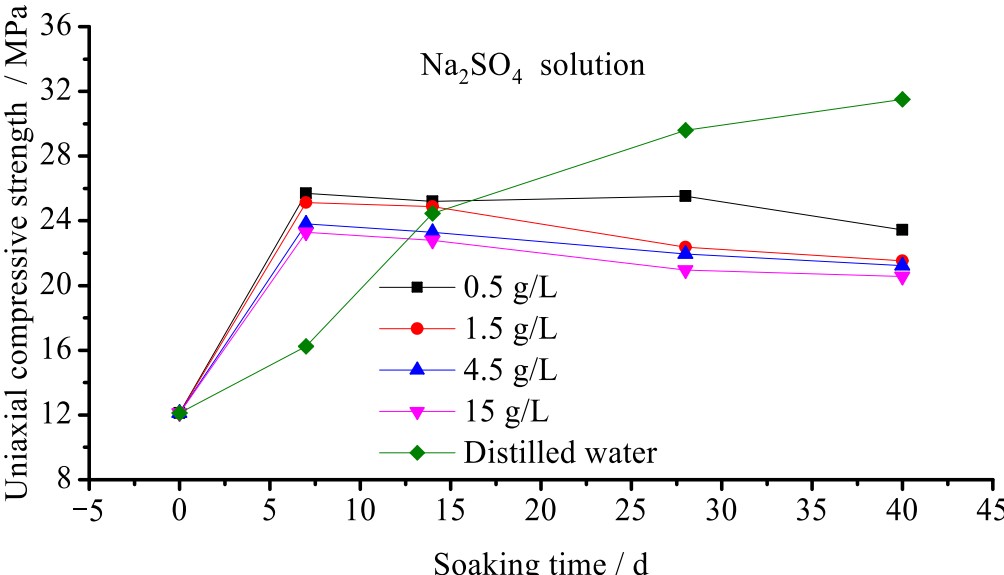

**Figure 3.** Uniaxial compressive strength of paste backfill specimens submerged in various sulfate solutions according to age.

It can be seen from Figure 3 that the uniaxial compressive strengths of the specimens submerged in the sulfate solution increased rapidly, but then slowly decreased, and the uniaxial compressive strengths of the specimens submerged in the low-concentration solution were greater than those submerged in the high-concentration solution. The uniaxial compressive strength of the specimen submerged in distilled water for 40 days is far greater than the uniaxial compressive strength of the specimen submerged in the sulfate solution for the same period of time. Combined with the observed changes in the appearances of the specimens, the reason for the rapid increase in the uniaxial compressive strength of the specimens in the early stage may be that the $SO_4^{2-}$ ions in the solutions infiltrated the specimens to participate in the hydration process. The formation of certain substances that fill in the pore structure of the specimens submerged in the sulfate solution caused their uniaxial compressive strengths to increase faster than that of the specimen submerged in distilled water. The uniaxial compressive strengths of the specimens submerged in the sulfate solution then began to decrease, likely for the same reason as the specimens submerged in the chloride solution: more and more hydration products were formed, gradually filling the pore structure and expanding to press against the walls, inducing tensile stress and causing internal damage that decreased strength.

Combined with Figures 2 and 3, the effect of chloride and sulfate on the compressive strength of the specimens is analyzed as follows. On 7 d, compared with the specimen in distilled water, the uniaxial compressive strength of the specimens in chloride and sulfate solution were greatly improved. This is also the reason why chloride and sulfate can be used as early strength agents of paste backfill. On 14 d, the uniaxial compressive strength of specimens in chloride solution was still increasing, while the uniaxial compressive strength of specimens in sulfate solution had begun to decrease. On 28 d, the uniaxial compressive strength of all specimens in sulfate solution was less than that in distilled water. However, the uniaxial compressive strength of specimens in 0.5 g/L, 1.5 g/L, and 4.5 g/L chloride solutions was still higher than that in distilled water, and only the uniaxial compressive strength of specimen in 15 g/L chloride solution is lower than that in distilled water. This result corresponds very well to the surface damage degree of the pattern in Figure 1. On 40 d, the uniaxial compressive strength of specimens in sulfate solution tended to be stable, and the strength difference between specimens in sulfate solution and distilled water further increased. This phenomenon indicates that the original structure of the specimen in sulfate solution has been destroyed. The uniaxial compressive strength of specimens

in 1.5 g/L, 4.5 g/L, and 15 g/L chloride solution have also begun to decline to a level lower than that of specimens in distilled water. On 40 d, although the uniaxial compressive strength of the specimen in chloride solution has not stabilized, we can still infer that its change trend is consistent with that in sulfate solution. That is, in the early stage of the test, chloride and sulfate significantly increased the uniaxial compressive strength of the specimen at a very fast speed. Then, with the gradual corrosion, the original structure of the specimen was destroyed and its uniaxial compressive strength decreased. Compared with sulfate, chloride has a longer strengthening time and higher strengthening effectiveness. This may be because the hydrophilicity of the material produced by the reaction of $SO_4^{2-}$ with the paste backfill was greater than that of the material produced by the combination of $Cl^-$ with the hydration products of the paste backfill, thus the expansion capacity of the former after being bound to water was much greater than that of the latter.

### 3.3. Microstructural Analysis

The macroscopic mechanical properties of a paste backfill specimen are actually determined by the material properties of the backfill and any change in its internal microstructure. The change in the compressive strength of the paste backfill when subjected to a corrosive environment is caused by changes in its internal material composition and microstructure under the action of $Cl^-$ and $SO_4^{2-}$. In order to fully understand the influence of the chloride and sulfate solutions on the mineral composition and microstructure of the backfill, X-ray diffraction (XRD) was used to conduct phase analyses of selected specimens. Figure 4 shows the results of an XRD analysis of the paste backfill specimen submerged in distilled water for 40 days.

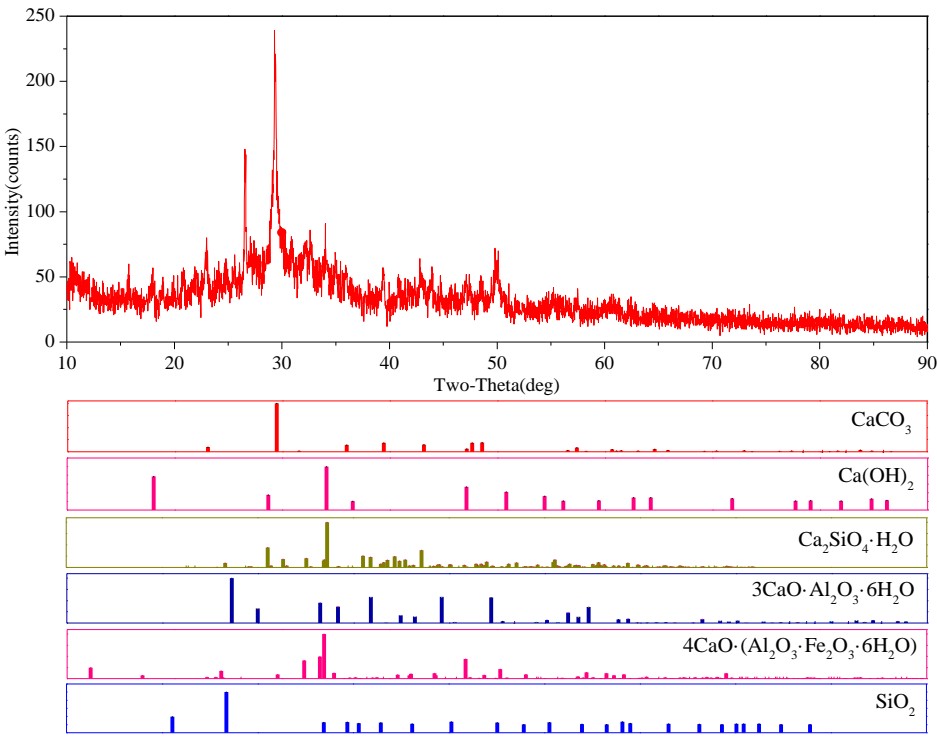

**Figure 4.** Diffraction pattern and main phases of paste backfill specimen submerged in distilled water for 40 days.

$SiO_2$ was the main component of fly ash and cement clinker in the paste backfill considered in this study. Both C-H and C-S-H are hydration products of the cement present in the paste, and are produced by the hydration reaction of clinker minerals as follows [42]:

$$2C_3S + H_2O = 3C\text{-}S\text{-}H + 3CH, \tag{1}$$

$$2C_2S + 4H_2O = 3C\text{-}S\text{-}H + CH. \tag{2}$$

Among these hydration products, there was a notably large concentration of C-S-H, which has a large specific surface area and rigid gel characteristics. It is difficult to dissolve C-S-H in water, where it forms amorphous colloidal particles owing to the van der Waals forces and chemical bonds between the gel particles. Furthermore, C-S-H has a high strength and is the primary source of the compressive strength of the paste backfill specimen. Under the SEM, it exhibits spherical, floc-like protrusions, smooth edges, compact structure, and no obvious sharp edges or corners, as shown in Figure 5. The CH content was also high, but as it has a layered structure with weak interlayer bonding and low strength, it was considered to contribute less to the observed specimen characteristics. Additionally, $3CaO\cdot Al_2O_3\cdot 6H_2O$, a stable hydration product of $3CaO\cdot Al_2O_3$, was present, but its crystals have poor mutual adhesion and low strength as well. Finally, $4CaO(Al_2O_3\cdot Fe_2O_3\cdot 6H_2O)$ was present as the hydration product of $4CaO\cdot Al_2O_3\cdot Fe_2O_3$, but it is extremely unstable and is considerably affected by temperature and the concentration of calcium hydroxide in the solution, as well as its ratio to $Al_2O_3/Fe_2O_3$.

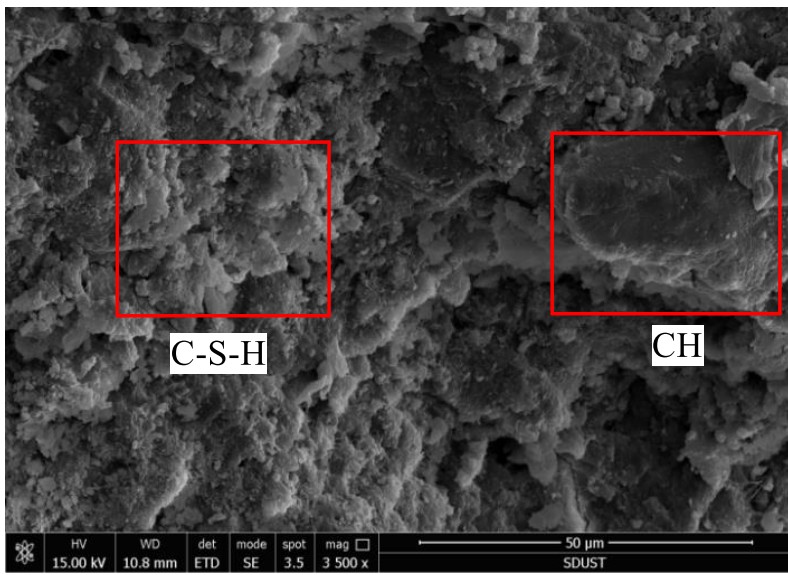

**Figure 5.** SEM image showing typical microstructure of backfill.

Figure 6 shows the results of an XRD analysis of the paste backfill specimen submerged for 40 days in a 15 g/L chloride solution. Compared with the results of the XRD analysis of the specimen submerged in a distilled water environment, the specimens submerged in the chloride environment lack hydrated calcium aluminate ferrite ($4CaO(Al_2O_3\cdot Fe_2O_3\cdot 6H_2O)$), which has more hydration reactivity than calcium chloroaluminate ($Ca_4Al_2O_6Cl_2\cdot 10H_2O$).

As calcium ferroaluminate hydrate is unstable, when it is constantly diluted by water it will be completely decomposed into $Ca(OH)_2$, $Al_2O_3$ aq, and $Fe_2O_3$ aq (where aq represents multiple water molecules), so when the specimen is moved from its natural hydration environment to be submerged in the chloride solution, it is continuously decomposed under the action of the large amount of water, causing the phase-hydrated calcium ferroaluminate to be missing. However, owing to the unstable nature of hydrated calcium ferroaluminate, its impact on the compressive strength of the backfill specimen is very small and can be ignored. Therefore, the changes in the strengths of the paste backfill specimens were determined by the presence of hydrated calcium chloroaluminate, produced by the reaction of $Cl^-$ with the components in the paste backfill, expressed as:

$$2Cl^- + Ca(OH)_2 = CaCl_2 + 2OH^-, \tag{3}$$

$$CaCl_2 + 3CaO\cdot Al_2O_3\cdot 6H_2O + 4H_2O = 3CaO\cdot Al_2O_3\cdot CaCl_2\cdot 10H_2O. \tag{4}$$

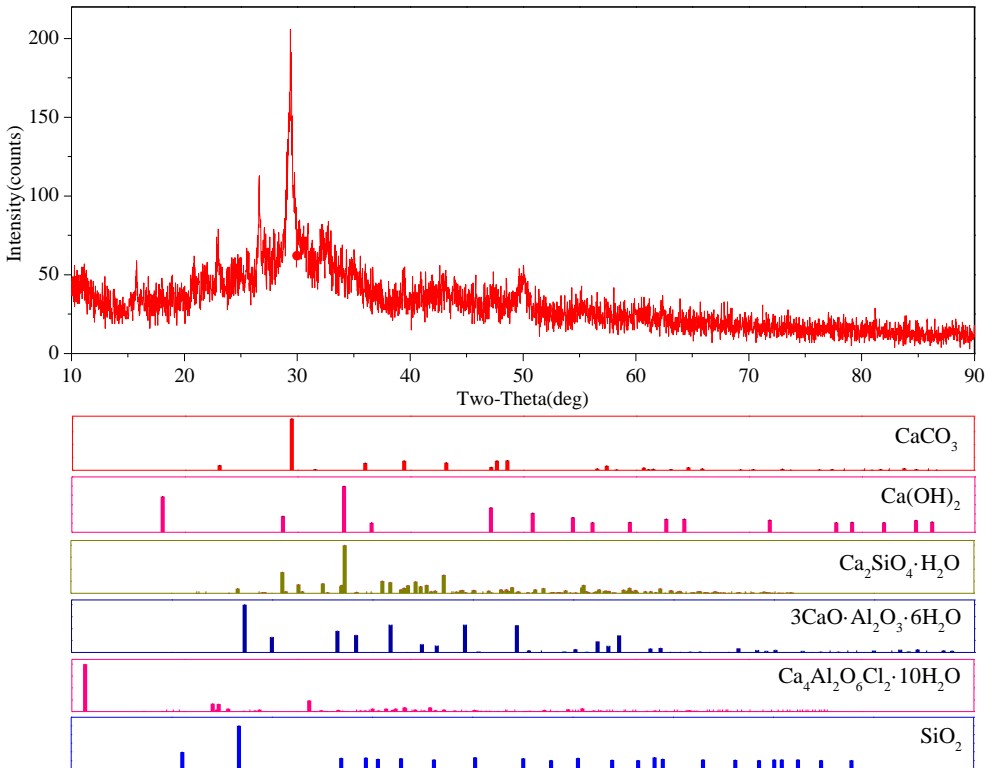

**Figure 6.** Diffraction pattern and main phases of paste backfill specimen submerged in an 15 g/L chloride solution for 40 days.

That is, the $Cl^-$ that penetrated into the backfill first reacted with $Ca(OH)_2$ to form $CaCl_2$, then this $CaCl_2$ reacted with a cement hydration product, tricalcium aluminate hydrate ($3CaO \cdot Al_2O_3 \cdot 6H_2O$), to form hydrated aluminum chloride calcium acid ($3CaO \cdot Al_2O_3 \cdot CaCl_2 \cdot 10H_2O$). Calcium chloroaluminate hydrate is an expansive material that filled pores and increased the compactness of the backfill in the initial stage of $Cl^-$ corrosion, causing the early strength of the backfill to develop faster, but in the later stage of corrosion, $3CaO \cdot Al_2O_3 \cdot CaCl_2 \cdot 10H_2O$ was produced, inducing tensile stress owing to expansion followed by tensile failure, internal damage, and microcracks that reduced the macroscopic compressive strength, as shown in Figure 7. The width of microcracks is generally about 10 μm.

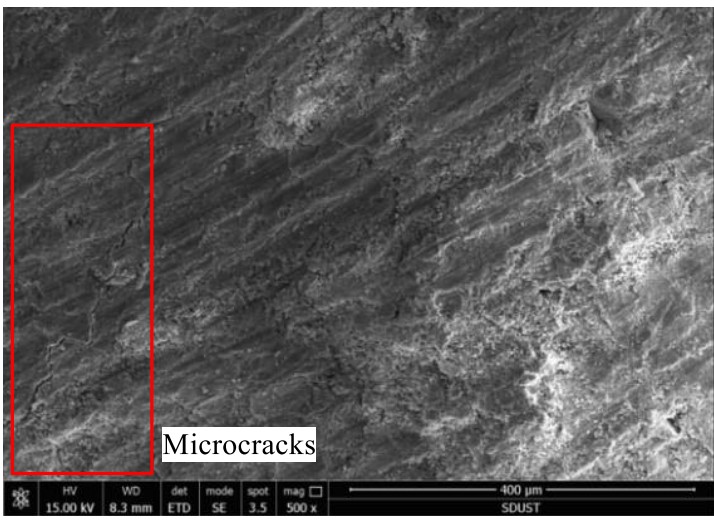

**Figure 7.** SEM image showing microcracks.

Figure 8 shows the results of an XRD analysis of the paste backfill specimen submerged for 40 days in a 15 g/L sulfate solution. Compared with the XRD analysis of the specimen submerged in a distilled water environment, more gypsum ($Ca(SO_4)\cdot2H_2O$) and ettringite ($Ca_6(Al(OH)_6)2(SO_4)_3\cdot26H_2O$) were observed in the sulfate-exposed specimen, and the presence of Glauber's salt ($Na_2SO_4\cdot10H_2O$) was indicated.

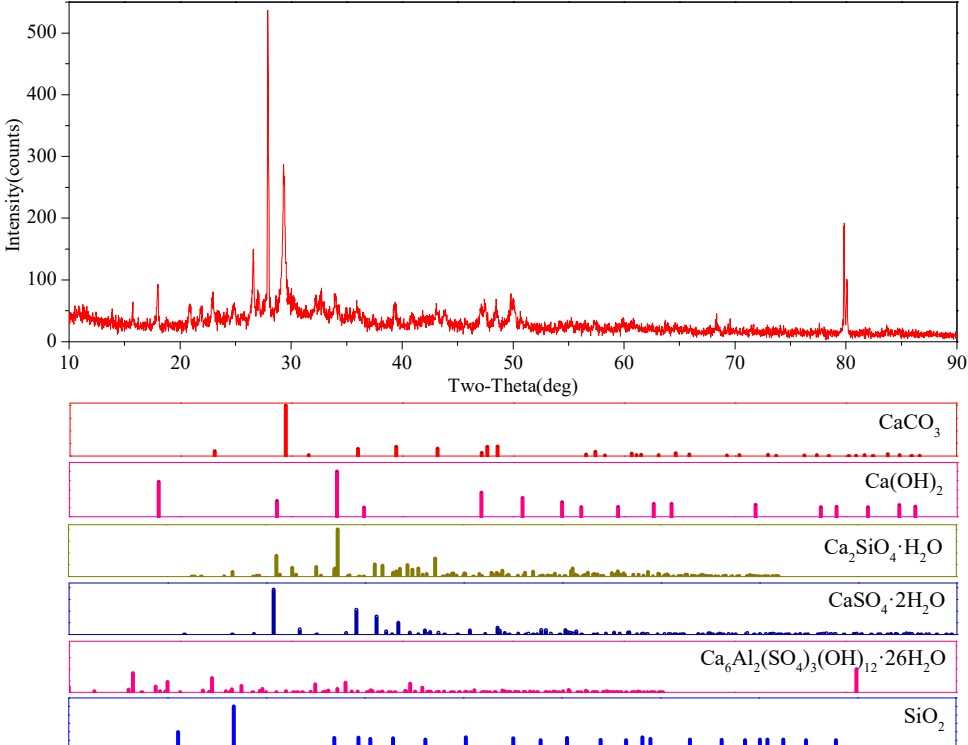

**Figure 8.** Diffraction pattern and main phase of paste backfill specimen submerged in a 15 g/L sulfate solution for 40 days.

The interaction of sodium sulfate and free calcium hydroxide produces gypsum with very little solubility. Because the volume of gypsum is much larger than that of the original compound, its formation may cause expansion and destruction of the paste backfill. The reaction process of sodium sulfate and calcium hydroxide to produce gypsum is

$$Na_2SO_4 + Ca(OH)_2 + 2H_2O = CaSO_4\cdot2H_2O + 2NaOH. \tag{5}$$

This gypsum-generating reaction can have two negative effects. The first is that gypsum precipitates and crystallizes in the pores of the backfill to form dihydrate gypsum. Owing to the expansive property of dihydrate gypsum, it will produce a great deal of tensile stress inside the backfill. When this stress is higher than the tensile strength of the paste, it will cause the backfill to expand and crack. The residual stress in the specimen also causes the slight deviation of the diffraction peak of gypsum (Figure 8) [43,44]. The second negative effect is that the C-S-H generated in the process of cement hydration is the formation of silicate crystal existing in the hardened backfill. The formation of silicate stone is a reversible process, so it serves as the basis for the stable existence of hydration products such as C-S-H. When sulfate reacts with cement hydration products to form gypsum, a large amount of $Ca(OH)_2$ will be consumed, causing the hydrolysis of hydrated C-S-H, which will lead to the loss of backfill material strength and durability.

There are two processes for the formation of ettringite, which was observed in notable quantities in the specimens submerged in the sulfate solutions. One is the direct reaction between gypsum and tricalcium aluminate or between gypsum and calcium aluminate hydrate to produce ettringite; the other is the reaction of tricalcium aluminate ($C_3A$)

with calcium to form aluminite ($C_6AS_3H_{32}$), which reacts to form monosulfur calcium sulfoaluminate ($C_4ASH_{12}$), which reacts with gypsum to produce ettringite. These ettringite formation methods are respectively called the direct reaction formation and indirect reaction formation, and are given as follows [45]:

$$\text{Direct reaction formation}: \begin{cases} C_3A + 3CSH_2 + 26H = C_6AS_3H_{32} \\ C_4AH_{13} + 3CSH_2 + 14H = C_6AS_3H_{32} + CH \end{cases} \quad (6)$$

$$\text{Indirect reaction formation}: \begin{cases} C_6AS_3H_{32} + 2C_3A + 4H = 3C_4ASH_{12} \\ C_4ASH_{12} + 2CSH_2 + 16H = C_6AS_3H_{32} \end{cases} \quad (7)$$

The fine needle-like or flaky ettringite crystals (hedgehog crystal, shown in Figure 9a) formed on the surface of the original aluminum-containing solid phase in the pores of the backfill absorbed water and expanded. The expanding ettringite crystals induced internal stress in the backfill, leading to cracking and destruction. This cracking made it easier for sulfate ions to penetrate into the interior of the backfill material, generating a vicious cycle that resulted in numerous cracks on the surface of the backfill specimen and the reduction of compressive strength. Thus, it was determined that the $C_3A$ in the specimen body served as the primary condition for the formation of ettringite: if the content of $C_3A$ is limited, the formation of ettringite will be inhibited. Therefore, controlling the relative content of $C_3A$ minerals in the paste backfill mix can improve its sulfate resistance.

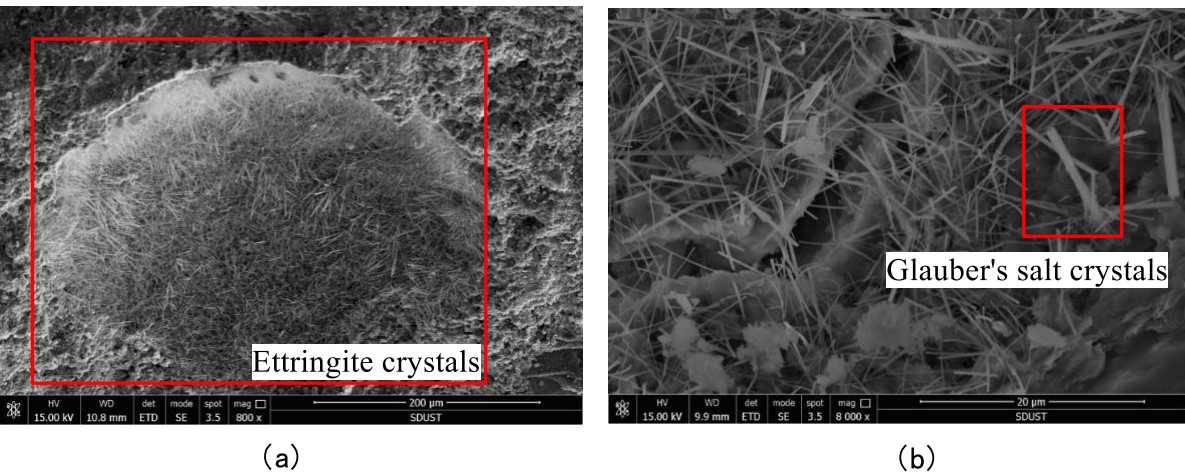

**Figure 9.** SEM images showing ettringite and mirabilite. (**a**) Ettringite crystals marked by red box. (**b**) Glauber's salt crystals marked by red box.

Glauber's salt crystals are short columnar or needle-shaped crystals, shown in Figure 9b. Salt crystal corrosion was observed in this study to be the most severe when the paste backfill specimens were submerged in sulfate solutions. Notably, $Na_2SO_4$ precipitates crystals above 32.3 °C as anhydrous sodium sulfate and below 32.3 °C as Glauber's salt $Na_2SO_4 \cdot 10H_2O$, for which the expansion coefficient is as high as 311. Thus, the formation of Glauber's salt will induce considerable expansion stress in the backfill, effectively destroying it.

Carbon particles were also observed in SEM. They have a porous structure as shown in Figure 10a. These carbon particles were introduced into the backfill material by the fly ash. They can only provide less support strength to the backfill [46], but their presence provides growth space for expansive substances, such as ettringite, mirabilite, etc., as shown in Figure 10b. Thus, when an expansive material is able to fill in the space among the carbon particles, the strength of the backfill will gradually increase; after filling these spaces, the expansive material will continue to expand and cause the backfill to undergo tensile failure, decreasing its compressive strength. Due to the low carbon content of coal gangue, whether there is a similar phenomenon needs to be further studied.

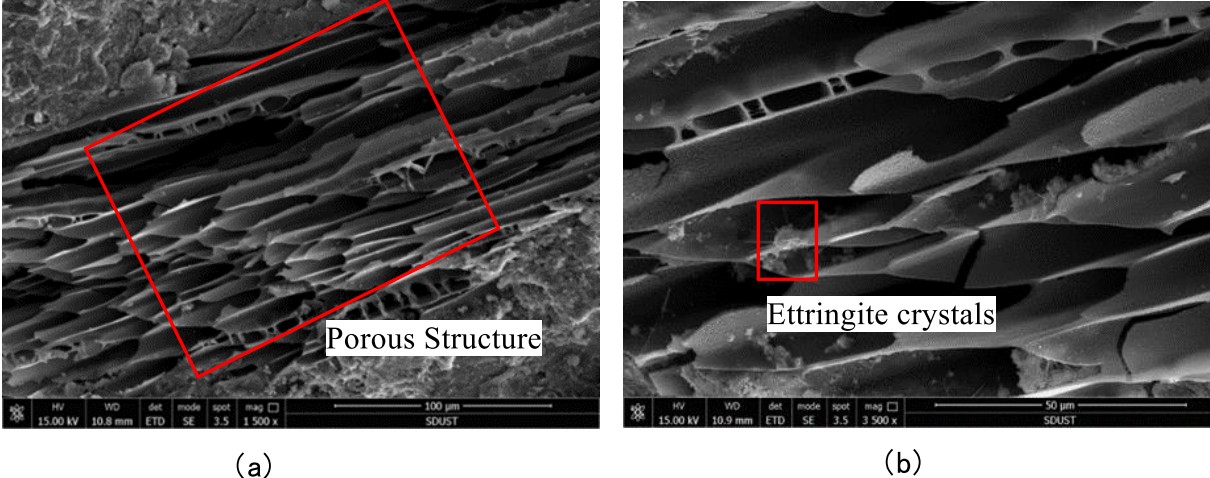

**Figure 10.** SEM images showing ettringite crystal formation among carbon particles. (**a**) porous structure of carbon particle marked by red box. (**b**) ettringite crystal marked by red box.

### 4. Conclusions

1. The presence of a chloride solution was found to improve the early strength of paste backfill because $Cl^-$ participates in the hydration reaction to form calcium chloroaluminate hydrate, which fills the internal pores of the backfill and improves its compactness. After a certain period of time, all internal pores are filled. Because the generated calcium chloroaluminate hydrate is expansive, damage is induced in the backfill material, causing it to gradually crack and ultimately resulting in a reduction in its compressive strength.

2. The presence of a sulfate solution was found to, on the one hand, corrode the backfill to produce dihydrate gypsum, ettringite, Glauber's salt, and other expansive substances, resulting in expansive stress inside the backfill. When this expansive stress exceeds the tensile strength of the backfill, internal damage occurs that gradually destroys the backfill material. This is referred to as crystalline expansion damage. On the other hand, the formation of gypsum, ettringite, etc. was found to consume $Ca(OH)_2$ during the process of filling the pores in the backfill, but $Ca(OH)_2$ serves as the basis for the stable existence of hydration products such as C-S-H. The reduction of $Ca(OH)_2$ will therefore inevitably enhance the decomposition of C-S-H, thereby reducing the internal bond strength of the backfill. This is referred to as reversible reaction damage. The strength and durability of paste backfill subjected to a sulfate solution will decrease under the action of these two types of damage.

3. Scanning the backfill specimens using SEM revealed that there were fine carbon particles in the backfill, introduced by the fly ash. Though the porous structure of carbon particles provides space for hydration products, helping to densify the backfill as it cures, it also provides space for the growth of expansive substances, reducing the strength of the backfill.

**Author Contributions:** Conceptualization, G.X.; Project administration, K.W.; Supervision, J.N.; Writing—original draft, G.X.; Writing—review & editing, K.F., K.W. and J.N. All authors have read and agreed to the published version of the manuscript.

**Funding:** This research was funded by Shandong Provincial Natural Science Foundation of China, grant number ZR2020QE136.

**Institutional Review Board Statement:** Not applicable.

**Informed Consent Statement:** Not applicable.

**Data Availability Statement:** Not applicable.

**Acknowledgments:** Thanks to the anonymous reviewers for constructive and enlightening comments and suggestions in the revision process.

**Conflicts of Interest:** The authors declare no conflict of interest.

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
