# Peer review of "Paste Backfill Corrosion Mechanisms in Chloride and Sulfate Environments"

_minerals, doi:10.3390/min12050551_

Round 1

Reviewer 1 Report

Paste Backfill Corrosion Mechanisms in Chloride and Sulfate Environments, by Guangzheng Xu et al.

This paper focuses on the dependency of compressive strength of cementitious pastes with respect to chlorides and sulfates chemical attacks. The studied material, which is composed of OPC, fly ash and a coal gangue, can be used as filing purposes in coal mines. The pastes were soaked in distilled water, in NaCl solutions and in Na2SO4 solutions with concentrations ranging from 0.5 to 15 g/L during 40 days. Uniaxial compressive cell, as well as XRD and SEM analysis, were performed. The NaCl and Na2SO4 solutions induced the formation of expansive hydrates (Friedel and Glauber salts, gypsum and ettringite) within the microstructure, leading to a mechanical reinforcement during the first weeks, but following by significant mechanical failure and cracks on the longer term.

Globally, all these processes are well-known in the literature from the point of view of the mechanical behavior and the chemical processes at stake. However, there is a clear interest to acquire a complementary set of data for paste backfills representative of industrial and practical issues in mines. Other aspects as the precipitation of expansive phases in the porosity of carbon particles from the fly ash are also interesting. I would recommend moderate revisions before publication in Minerals.

Note. In the comments below, Cl stands for chlorides and SO4 for sulfates.

C1. Mine water chemistry, in the Introduction.

Some information on the range of Cl and SO4 concentrations usually encountered in Chinese mine waters should be added for comparison with the concentrations used in the present tests (0.5 g/L to 15 g/L). It should also be useful to provides (in the Introduction or materials section) for the concentrations in mol/L in addition to g/L for comparison with literature studies expressed in mol/L units.

C2. Several key initial parameters of the pastes are lacking, in Section 2.

The proportions in wt.% of OPC, fly ash and coal gangue components are lacking. The relative contents between OPC and fly ash will impact the resistance of the cement paste backfill with respect to the chemical attacks (portlandite content and C3A content vs. secondary ettringite formation…). Is this specific tested backfill material (recipes) already used in Chinese mines?

The water on cement/solid ratio (W/C or W/S) is lacking. It would impact the mechanical resistance of the paste backfill with respect to the chemical attacks and mechanic resistance (diffusion coefficient, microstructure).

The initial microporosity of the paste is lacking. Different porosities could change the mechanical effect of expansive phase precipitation.

The size/dimensions and morphology (cylindric…) of the pastes/blocks are lacking. Furthermore, did the compressive cells impose lateral containment constraints or was it a free deformation/strain test?

The pastes were not cured but directly soaked in the chemical solutions. Was it to mimic what occurs in real mines?

C3. Long time evolutions of compressive strengths in Cl and SO4 soaking solutions, in Section 3.1.

After 40 days, there was a weaker impact of Cl compared to SO4. But on the other hand, the trend (slope) is still decreasing after 40 days in the case of Cl (Fig. 2), compared to SO4 case (Fig. 3) in which a plateau is reached after 30 days. This main difference at 40 days – plateau (SO4) vs. decreasing trend (Cl) – should be discussed at minima in the text.

Furthermore, why were the CL experiments stopped after 40 days instead of continuing them up to reach a stabilized state? What did justify such a duration of 40 days? Was it a national standard? Was it due to the high cracking state of the paste after 40 days?

C4. Important information lack in the figures and discussion of the XRD spectra, in Section 3.3.

The term “X days” is written both in the captions of Figs. 4, 6 and 8 and in the text (lines 174, 196, 229). What does it mean? X = 40 days, the longer leaching time?

The same for “X g/L” in both captions of Figs. 6 and 8 and in the text (lines 196, 229). To which concentration do the XRD spectra correspond? The highest one, i.e. 15 g/L?

Along the same lines, it would be interesting to discuss of the XRD spectra to compare the quantities/amounts of Ca-chloroaluminate) that formed at 0.5 g/L and at 15 g/L in the Cl tests. Same recommendation for gypsum and ettringite quantities in the SO4 tests.

C5. Crack network/pattern, in Fig. 7.

Any information on the crack network (density of cracks, mean aperture) could be added in the discussion.

C6. Fig. 8 seems wrong.

Fig. 8 is exactly the same one as for Cl tests (Fig. 6), not the one for SO4 tests. This important mistake should be corrected. The reviewers (and readers) had to take for granted what the authors discussed in the text.

C7. Precipitation of expansive phases in the porosity of carbon particles from the flay ash, in Fig. 10.

It would be interesting to mention in the text whether a similar process took place or not in the coal gangue.

C8. CH phase and silicone, in lines 245 – 247.

I do not understand the meaning of that sentence: “Ca(OH)2 generated in the process of cement hydration is itself made of silicon”. What is a “Ca(OH)2 phase made of silicon”? This should be clarified.

C9. Spelling errors.

Line 56 “Cl- chloride”, delete ‘Cl-“

Eq. 3. It should be “2 Cl-“ and “2 OH-“ in the stoichiometric equation.

Eq. 7. Suppress Eq. 7-1, which is identical to Eq. 6? Eqs. 7-2 and 7-3 seem correct.

Line 304. Would the terms “reduce the decomposition” rather be “enhance the decomposition”? Please check.

Author Response

Dear Reviewer:

Thank you for your comments concerning our manuscript entitled “Paste Backfill Corrosion Mechanisms in Chloride and Sulfate Environments”. Those comments are all valuable and very helpful for revising and improving our paper. According to your comments, we have made careful corrections which we hope meet with approval and mark up in Word. Please see the attachment. Thank you very much.

Sincerely,

Guangzheng Xu, Kegong Fan, Kun Wang(Corresponding author),Jianguo Ning(Corresponding author)

College of Energy and Mining Engineering, Shandong University of Science and Technology

wkun@sdust.edu.cn(K.W.); njglxh@126.com(J.N.);

Reviewer 2 Report

This is a meaningful paper. The content of the paper is very rich and the explanation is relatively comprehensive. Before being accepted, I hope the author can clarify the following questions:

  1. In lines 289 and 290, the conclusion. The author points out the formation of calcium chloroamine hydrate. It is hoped that the formation process of calcium chloroamine hydrate can be proved by chemical reaction equation and microscopic test results.
  2. In lines 173 and 235, would you like to explain what "x days" means?
  3. In order to make the introduction more complete, it is suggested that the authors also refer to the following articles:

[1]        Jingwei Ying, Zhijun Jiang, Jianzhuang Xiao. Synergistic effects of three-dimensional graphene and silica fume on mechanical and chloride diffusion properties of hardened cement paste [J]. Construction and Building Materials, 2022, 316(17): 1-12.

[2]        Chen Shaojie, Du Zhaowen, Zhang Zhen, et al. Effects of chloride on the early mechanical properties and microstructure of gangue-cemented paste backfill [J]. Construction and Building Materials, 2020, 235

[3]        Jiang Haiqiang, Fall Mamadou. Yield stress and strength of saline cemented tailings materials in sub-zero environments: slag-paste backfill [J]. Journal of Sustainable Cement-Based Materials, 2017, 6(5): 314-331.

Author Response

(The authors gave the same response as above.)

Reviewer 3 Report

In this study, the effect of chloride and sulfate solutions on strength of paste backfill and microstructural formations in this process were evaluated. My comments and suggestions in this context are given below.

  1. Could a numerical expression describing the conclusion of the article be given in the abstract?
  2. Line 12-22: Given as general information about the effects of chloride and sulfate in cementitious mixtures... It might be better to use statements that emphasize the purpose and results of the study.
  3. Line 47-48: “However, owing to the complexity of SO42- corrosion, there remain many corrosion problems that have not been clarified.” What are the many corrosion problems here, and which ones are described in this study differ from other studies?
  4. Line 61-64: What are the physical and chemical properties of cement, fly ash, and coal gangue materials used in the mixture? Also, what is the mixing ratio? How were the binder mixing ratio and the desired strength value chosen? Why did fly ash join?
  5. Line 10 and Line 67: “.....sulfate solutions in concentrations of 0 g/L, 0.5 g/L, 1.5 g/L, 4.5 g/L, or 15 g/L for 7, 14, 28, and 40 days,..” How were these rates and days chosen? How much of the solution is chloride and how much is a sulfate? Or separately? It should be explained.
  6. Line 83 and 86: “....for 30 days in different concentrations of chloride and sulfate solutions....” 30 days not previously specified herein expire. Why is that?
  7. It has been observed that there is no discussion for the results in Figure 1, Figure 2, and Figure 3 compared to the results of different or similar studies.
  8. Line 173 and 175 and 204: “.... for X days.” What is X here?
  9. Line 203 and 229: X g/L...?

Author Response

(The authors gave the same response as above.)

Round 2

Reviewer 1 Report

The authors have carefully and cleverly taken into account most of my comments. Thanks. I now strongly recommend publication in Minerals.

Author Response

Dear Reviewer:

Thank you very much.

Sincerely,

Guangzheng Xu, Kegong Fan, Kun Wang(Corresponding author),Jianguo Ning(Corresponding author)

College of Energy and Mining Engineering, Shandong University of Science and Technology

wkun@sdust.edu.cn(K.W.); njglxh@126.com(J.N.);